# Kathryn V. Holmes: A Career of Contributions to the Coronavirus Field

**DOI:** 10.3390/v14071573

**Published:** 2022-07-20

**Authors:** Aurelio Bonavia, Samuel R. Dominguez, Gabriela Dveksler, Sara Gagneten, Megan Howard, Scott Jeffers, Zhaohui Qian, Mary Kathryn Smith, Larissa B. Thackray, Dina B. Tresnan, David E. Wentworth, David R. Wessner, Richard K. Williams, Tanya A. Miura

**Affiliations:** 1Vaccine Development, Bill & Melinda Gates Medical Research Institute, Cambridge, MA 02139, USA; aurelio.bonavia@gatesmri.org; 2Department of Pediatrics-Infectious Diseases, University of Colorado School of Medicine, Aurora, CO 80045, USA; samuel.dominguez@childrenscolorado.org; 3Department of Pathology, Uniformed Services University of the Health Sciences, Bethesda, MD 20814, USA; gabriela.dveksler@usuhs.edu; 4Division of Viral Products, Office of Vaccines Research and Review, Center for Biologics Evaluation and Research, Food and Drug Administration, Silver Spring, MD 20993, USA; sara.gagneten@fda.hhs.gov; 5Battelle Memorial Institute, Columbus, OH 43201, USA; megan.w.howard@gmail.com; 6GenSight Biologics, 75012 Paris, France; sjeffers@gensight-biologics.com; 7Institute of Pathogen Biology, Chinese Academy of Medical Sciences, Beijing 100050, China; zqian2013@sina.com; 8General Dynamics Information Technology, Falls Church, VA 22030, USA; marykathryn.smith@gdit.com; 9Department of Medicine, Washington University School of Medicine, St. Louis, MO 63110, USA; lthackray@wustl.edu; 10Safety Surveillance and Risk Management, Worldwide Safety, Pfizer, Groton, CT 06340, USA; dina.b.tresnan@pfizer.com; 11COVID-19 Emergency Response, Virology Surveillance and Diagnosis Branch, Influenza Division, Centers for Disease Control and Prevention, Atlanta, GA 30329-4027, USA; gll9@cdc.gov; 12Departments of Biology and Public Health, Davidson College, Davidson, NC 28035, USA; dawessner@davidson.edu; 13NIAID, NIH, Bethesda, MD 20817, USA; rick6005@gmail.com; 14Department of Biological Sciences, University of Idaho, Moscow, ID 83844, USA

**Keywords:** coronavirus, coronavirus receptors, coronavirus spike glycoprotein

## Abstract

Over the past two years, scientific research has moved at an unprecedented rate in response to the COVID-19 pandemic. The rapid development of effective vaccines and therapeutics would not have been possible without extensive background knowledge on coronaviruses developed over decades by researchers, including Kathryn (Kay) Holmes. Kay’s research team discovered the first coronavirus receptors for mouse hepatitis virus and human coronavirus 229E and contributed a wealth of information on coronaviral spike glycoproteins and receptor interactions that are critical determinants of host and tissue specificity. She collaborated with several research laboratories to contribute knowledge in additional areas, including coronaviral pathogenesis, epidemiology, and evolution. Throughout her career, Kay was an extremely dedicated and thoughtful mentor to numerous graduate students and post-doctoral fellows. This article provides a review of her contributions to the coronavirus field and her exemplary mentoring.

## 1. Introduction

Over the course of a research career spanning more than five decades, Kathryn (Kay) Holmes has contributed a wealth of critical knowledge to the field of virology. From her Ph.D. work at the Rockefeller University with Purnell Choppin studying cell fusion by simian virus 5, to identifying lymphocytic choriomeningitis virus as a cause of fatal callitrichid hepatitis in primates, to starting a company focused on the development of universal influenza vaccines, Kay has always been fascinated by a diversity of viruses. However, most of her research has focused on coronaviruses. Her interests in coronaviruses have included host and tissue specificity, viral spike/receptor interactions, viral and cell fusion mechanisms, pathogenesis, and epidemiology. Here, we focus on Kay’s contributions to the coronavirus field, which have provided a foundation for current research on and development of vaccines and therapies for pandemic coronaviruses. In addition to her remarkable scientific contributions, one of her most influential activities has been mentoring numerous graduate students and post-doctoral researchers. The authors of this article are a subset of those trainees and are forever grateful for Kay’s enthusiastic support and dedication to launching and supporting our interests in virology and our careers.

## 2. Characterizing Coronaviral Structural Proteins

In 1977, while at the Uniformed Services University of the Health Sciences, Kay first published work on coronaviruses with Larry Sturman, who had begun to characterize the structural proteins of the murine coronavirus mouse hepatitis virus (MHV-A59) [1]. Initially, they identified and characterized the two membrane-associated envelope proteins of MHV-A59: the spike glycoprotein (S) that makes up the large petal-shaped peplomers on the envelope and the smaller, transmembrane matrix glycoprotein (M) important for virion morphogenesis and assembly [2,3]. They determined that, like many other viral glycoproteins, S was an N-linked glycoprotein. In contrast, they determined that M was an O-linked glycoprotein, potentially making it the first identified viral glycoprotein of this type [4]. Kay also studied the major coronaviral RNA binding protein, nucleocapsid (N), and its role in viral replication [5,6]. Later, Kay and her coauthors defined the role of S in virion attachment and cell-fusing activities [7]. They determined that S was proteolytically cleaved into two different domains, the N-terminal half (90B or S1) and the C-terminal half (90A or S2), during virion maturation or through exogenous protease treatment. They also discovered how the regulation of S cleavage, the location of the cleavage site, the rate of transport of cleaved S to the cell membrane, and the lipid composition of the host cell membrane determined the extent of MHV-A59-induced syncytia formation [7,8,9]. Finally, Kay and her coauthors elucidated how a temperature-dependent, alkaline pH-dependent, irreversible conformational change in S led to the shedding of S1 and viral fusion at the plasma membrane [10,11]. These efforts set the stage for Kay’s later work spanning multiple areas of coronavirus virology, including receptor discovery, further S protein characterization, pathogenesis, and epidemiology (Figure 1).

## 3. Discovering Coronavirus Receptors

### 3.1. CEACAM1a as a Receptor for MHV

When Kay’s lab identified the MHV-A59 receptor in 1991, only a few viral receptors, including those for HIV, rhinovirus, and poliovirus, had been identified [12,13,14]. As Kay correctly predicted for human and murine coronaviruses, the presence of virus-specific receptors and co-receptors was found to be an important determinant of viral host specificity and cellular tropism within a host [15]. Kay’s interesting observation that the SJL/J mouse strain was resistant to MHV-A59 infection suggested that these mice either lacked the receptor or had a mutated form of it. Confirming this supposition required the development of a creative technique, the virus overlay protein blot assay, which demonstrated MHV-A59 binding to a 110 kDa protein in cell membranes from susceptible, but not resistant, mouse strains [15]. Additional elegant experiments identified the cellular receptor for MHV-A59 as a member of the carcinoembryonic antigen (CEA) glycoprotein family, later known as CEACAM1a [16,17]. Additional strains of MHV, including DVIM that expresses a hemagglutinin esterase protein and JHM, were also shown to require CEACAM1a for infection [18]. Using several techniques, including RT-PCR with degenerate primers based on the N-terminal domain of the putative receptor glycoprotein in susceptible mouse strains and site-directed mutagenesis, Kay and her colleagues showed that MHV-A59 bound to the first Ig-domain of CEACAM1, formerly designated as biliary glycoprotein (BGP1) or MHV receptor (MHVR) [19,20,21,22]. In contrast, MHV-A59 bound with much lower affinity to an allelic variant of CEACAM1 expressed in the resistant SJL/J mouse strain and did not bind to brush border membranes from species other than mouse [23]. These were seminal results in virology that linked receptor binding to host species specificity. CEACAM1a-null mice were resistant to MHV-A59 infection and mice with impaired expression of the four Ig domain isomers of CEACAM1a had reduced susceptibility, further demonstrating the importance of this molecule for viral pathogenesis [24,25]. Interestingly, Kay and collaborators later showed that neurotropic MHV strain JHM could infect and spread in the brain of CEACAM1a-null mice, though less efficiently than in wild-type mice [26]. In total, this body of work represents one of the early molecular approaches to identifying and characterizing a viral receptor and has contributed significantly to our understanding of viral cell entry and pathogenesis.

### 3.2. APN as a Receptor for HCoV-229E and Other Alphacoronaviruses

Although human coronaviruses were isolated in the 1960s, researchers knew very little about the diversity and function of the human coronavirus S protein or the nature of its receptor even by 1990. The first breakthrough came in 1992 when Kay’s lab identified human aminopeptidase N (hAPN or CD13) as the receptor for human coronavirus 229E (HCoV-229E) [27]. A monoclonal antibody raised against plasma membranes of susceptible cell lines blocked infection by HCoV-229E. In addition, immunoprecipitation assays with this antibody led to identification of hAPN as the potential receptor. Confirmation came from molecular approaches, including the expression of hAPN in a non-permissive mouse cell line, which rendered these cells susceptible to HCoV-229E infection. Further characterization of this protein showed that the hAPN catalytic site was important for receptor activity, but its enzymatic activity was not required for infection. At the same time, porcine (p)APN was determined to be the receptor for the porcine coronavirus TGEV [28], but there was a clear species specificity. HCoV-229E could not use pAPN as its receptor and TGEV could not use hAPN as its receptor [29,30].

Kay’s lab members explored this species specificity further and demonstrated that HCoV-229E bound to membranes of feline, canine, porcine, and human cell lines, and intestinal brush border membranes from these species but only infected human and feline cells [30]. Of note, transfection of the various non-permissive cell lines with genomic RNA of HCoV-229E resulted in virus production. Thus, a step in the viral replication cycle between binding and transcription contributed to species specificity of HCoV-229E. Kay’s group also identified feline (f)APN as the receptor for the two feline coronaviruses FIPV and FeCoV [31]. Interestingly, they determined that fAPN was a universal receptor for the former group 1 coronaviruses, now alphacoronaviruses, including HCoV-229E, TGEV, and the canine coronavirus CCoV. Her lab members further demonstrated that introducing a single glycosylation site in hAPN prevented HCoV-229E infection [32]. However, deletion of the homologous glycosylation site in pAPN did not enable infection by HCoV-229E. These data suggested a role for additional determinants in the species specificity of HCoV-229E. To conduct pathogenesis studies with HCoV-229E in vivo, Kay’s laboratory engineered a transgenic mouse line expressing hAPN, and went on to show that although hAPN was expressed in cells from these transgenic mice and selected cells could be infected in vitro, the mice were resistant to infection in vivo, suggesting that other host factors were required for infection [33]. Additional molecular studies used mutational analysis of APN and the generation of murine/feline chimeras to show that three areas of APN were important for host range, but the determinants were not identical for all alphacoronaviruses [34].

In contrast to the other alphacoronaviruses, HCoV-NL63 was unable to use hAPN or fAPN as a receptor [35]. Surprisingly, Kay’s lab and others subsequently identified angiotensin converting enzyme 2 (ACE2) as an entry receptor for HCoV-NL63 [35,36]. ACE2 has been identified as a receptor for betacoronaviruses SARS-CoV and SARS-CoV-2 [37,38]. Thus, major shifts appear to have occurred in closely related coronaviruses such that they do not necessarily use related receptor proteins, and more distantly related coronaviruses convergently evolved to use the same receptor protein.

### 3.3. CD209L/L-SIGN as a Receptor for SARS-CoV

When SARS coronavirus (SARS-CoV) emerged in human populations in late 2002, Kay’s laboratory joined efforts by many investigators to characterize the new pandemic virus. Having identified CEACAM1 and APN as receptors for MHV-A59 and multiple alphacoronaviruses, respectively, her lab was uniquely situated to identify a receptor for SARS-CoV. Kay’s group transduced a cell line that was resistant to SARS-CoV infection with a retroviral vector expressing a human lung cDNA library and sorted cells based on SARS-CoV S binding [39]. These cells were used to identify CD209L/L-SIGN as a potential receptor or co-receptor for SARS-CoV entry. Although ACE2 has been shown to be the major receptor for both SARS-CoV and SARS-CoV-2 [37,38,40], other laboratories have reported that CD209L/L-SIGN also may serve as a receptor for SARS-CoV-2 [41].

## 4. Characterizing Spike: Receptor Interactions and Fusion Activity

In addition to identifying coronavirus receptors and determining their role in viral host range, Kay eagerly explored the molecular interactions between S proteins and their receptors. Starting with MHV-A59 and CEACAM1a, her lab characterized S/receptor interactions using mutational analysis of S and monoclonal antibody binding sites in CEACAM1 [42]. The finding that multiple different CEACAM proteins and isoforms functioned as receptors for MHV-A59 provided insight into receptor determinants of infection [43,44]. After extensive passage of persistently infected murine cells, Kay’s lab discovered a host-range mutant of MHV-A59, MHV/BHK, that was able to infect a wide range of non-murine cell lines [45]. Characterization of this mutant revealed that residue changes within the N-terminal domain of S changed its receptor specificity and viral host range [46,47,48].

Kay collaborated with structural biologists Jia-huai Wang (Harvard Medical School) and Fang Li (University of Minnesota) to solve crystal structures of the murine CEACAM1a protein, the N-terminal domain of MHV-A59 S in complex with CEACAM1a, and the bovine coronavirus S N-terminal domain [49,50,51]. These studies provided structural insight into MHV-A59 S/CEACAM1a and BCoV S/glycan interactions. Kay’s and Fang Li’s groups further solved the crystal structure of the murine CEACAM1b protein, revealing critical insight into the differences in MHV receptor activity of CEACAM1a and CEACAM1b [16,21,52].

Kay’s early work investigated the pH-triggering of MHV-A59 S with and without receptor binding and showed that receptor-independent syncytia formation resulted from a delicate balance between S protein stability and viral fitness [10,11]. Her group subsequently described conformational changes in MHV-A59 S triggered by either receptor binding or pH 8, showing that cleavage between S1 and S2 was not required for conformational changes associated with fusion [53]. Later work from her lab identified a histidine in the 209 position of MHV-A59 S protein as a key pH sensor for this process and showed that a single substitution (G29P) arrested S in a prefusion state, even when bound to its receptor [54,55]. These studies provided vital clues about the molecular mechanisms of S-mediated membrane fusion and viral entry.

After discovering the role of APN proteins as coronaviral receptors, Kay’s lab continued to characterize HCoV-229E S interactions with hAPN. Molecular approaches demonstrated that the S1 region between amino acids 417 and 547 harbored the receptor binding domain of HCoV-229E [56]. However, the initial steps for HCoV-229E entry remained unknown. To investigate this topic, Kay’s lab used drugs to inhibit endosomal acidification and showed that HCoV-229E entered cells via hAPN-dependent endocytosis [57]. Although there was no known cleavage event for the HCoV-229E S glycoprotein, conformational changes that occurred at 37 °C, but not at 4 °C, were observed in vitro using soluble receptor and S protein constructs coupled with neutralization assays [58].

Prior to the identification of a receptor for HCoV-HKU1, Kay’s group generated monoclonal antibodies and recombinant truncated S proteins to map receptor binding function to the C-domain of S1 [59]. This finding contrasted with other betacoronaviruses (MHV, OC43, and BCoV), whose S proteins were shown to bind to receptor proteins or sugars by the N-terminal domain of S1. Alphacoronaviruses, including HCoV-229E, TGEV, and HCoV-NL63, were also found to have receptor binding activity in the C-domain of S1. Thus, Kay and her colleagues provided a new example of the modular nature of CoV S proteins. Binding and entry of two CoVs in the same phylogenetic group could be initiated by different regions of the S glycoprotein.

Following receptor binding by the S1 domain, the C-terminal portion of S, the S2 domain, plays a major role in large conformational changes in S that mediate membrane fusion. In the early days of the SARS-CoV pandemic, Kay partnered with Robert Hodges (University of Colorado Health Sciences Center) to dissect the role S2 played in SARS-CoV infection. Their work resulted in biophysical characterization of the heptad repeats, the juxtamembrane domain, and the fusion peptide of the SARS-CoV S protein [60,61,62]. Locating and targeting the heptad repeats was essential for understanding S2 conformational changes that mediate membrane fusion and designing strategies to interrupt those changes [61,63,64]. In addition to investigating the fusion peptide of SARS-CoV, Kay’s group also identified the fusion peptides in the S glycoproteins of another emergent human CoV, MERS-CoV, and MHV-A59 [62]. Although the amino acid sequences of the fusion peptides of these divergent CoV S proteins were found to be quite different, they had conserved functions and locations within S. These foundational studies have provided critical insight into the ongoing development of therapies and vaccines that may be effective against the currently circulating SARS-CoV-2 virus and future pandemic strains.

## 5. Understanding Coronaviral Pathogenesis

To better understand disease pathogenesis, Kay applied the cellular and molecular details of CoV biology in murine and primary cell model systems. Much of this work focused on using MHV-A59 as a model for demyelinating disease. In a productive collaboration with Monique Dubois-Dalcq (NIH), Kay and her colleagues characterized cellular and molecular mechanisms of demyelination and remyelination in MHV-A59 infected mice. Their work associated differences in neuropathogenesis among MHV strains with differences in tropism, cytopathic effects, and virion assembly in neuronal versus non-neuronal cell types within cultured cells from murine spinal cords [65]. In multiple mouse strains, they characterized the pathology, kinetics, and location of viral RNAs and antigens and the expression of host genes through the process of MHV-A59-induced demyelination and remyelination [66,67,68,69,70]. In these studies, cell type-specificity was carefully evaluated, and the findings were confirmed using primary cell models [65,71]. This work not only provided critical mechanistic insight into viral demyelination but also into the mechanisms leading to remyelination and recovery from disease.

In addition to dissecting the roles of various cell types in viral demyelinating disease, Kay was also interested in cell-type specificity of CoV infections in the respiratory tract. Using polarized airway epithelial cells and human tracheal explanted tissues, her group showed that HCoV-229E entered and exited polarized airway cells apically [72]. Collaborating with Robert Mason (National Jewish Health), an expert in the isolation and culture of primary, differentiated alveolar epithelial cells, Kay’s group evaluated lung cell-type specificity and immune responses to infection by SARS-CoV, HCoV-229E, and rat coronaviruses [73,74,75,76,77,78]. With Peter Rottier’s group (Utrecht University), Kay and colleagues performed studies to correlate coronavirus entry and exit of polarized cells with infection and spread in the enteric tract. They found that MHV-A59 entered on the apical side but exited the cell from the basolateral side [79]. In contrast, TGEV entered and exited polarized cells from the apical surface [79]. These studies contributed to the knowledge of the cell-type specificity of respiratory CoV infections and established primary cell models to evaluate cell-type contributions to immunity and pathogenesis.

## 6. Characterizing Coronaviral Epidemiology

Following the SARS epidemic in 2003, renewed interest in human coronaviruses led to the discovery of HCoV-NL63 and HCoV-HKU1 [80,81]. Kay expanded her research into understanding the clinical and molecular epidemiology, pathophysiology, and disease associations of these previously unknown viruses. Her group’s work documented that HCoV-NL63 and HCoV-HKU1 caused significant respiratory disease in children with seasonal and yearly variations [82,83]. These studies also demonstrated that, surprisingly, the N-terminal domain of the HCoV-NL63 S protein was the most variable part of the genome and found evidence of recombination between strains [84]. In contrast, Kay’s group found remarkable sequence conservation in HCoV-HKU1 viruses circulating throughout the world [83].

In addition to this work on “new” human coronaviruses, Kay astutely recognized the role of zoonotic viruses as emerging pathogens. The recognition that SARS-CoV emerged from Asian bats highlighted the importance of bats as reservoir hosts for emerging viruses [85,86,87], and Kay began exploring the extent to which bats could harbor viruses and serve as potential reservoirs for disease outbreaks worldwide. Towards this end, her lab was the first to demonstrate that bats in North America harbored a diverse array of CoVs [88]. Further exploring the ecology of these viruses, her work suggested that the ongoing evolution of CoVs in bats would provide a continued threat of the emergence in new host species [89]. These studies found a high prevalence of alphacoronavirus RNAs in big brown bats in roosts in proximity to human habitations and known to have direct contact with people. These data suggested the significant potential for cross-species transmission of CoVs. Collaborating with others around the world, her lab also found novel CoVs in bats in Latin and South America [90]. Kay’s work in this area expanded knowledge of and the impetus to further study the role of bats in zoonotic viral outbreaks [87], which has been quite relevant to the current COVID-19 pandemic.

## 7. Mentoring Virologists

In addition to her many meaningful contributions to our understanding of coronaviruses, Kay also served as a dedicated and valued mentor. We recall our experiences not only in the details of virology that we learned, but also in more personal ways. Kay’s holistic approach to mentoring made working with and learning from her unique. We joined Kay’s lab because of her infectious curiosity and enthusiasm. She would read Science magazine while brushing her teeth to keep up on new articles and learn new things. Her joy spread to her mentees and empowered them to be curious about coronaviruses but also other areas of science. Kay’s encouragement led us to put aside fears of not-knowing, introduce ourselves to everyone, and to stay curious. We forged connections leading to collaborations, friendships, and job opportunities.

In Kay’s group, everyone worked on proposals and reviewed data. “Never apologize for your data”, a mantra many of us learned from Kay, was learned alongside thoughtful and elegant experimental design. Kay’s knowledge of techniques, applications of techniques across fields, and fearlessness to try new or out-of-the-box ideas in the lab created a generation of scientists unafraid to step outside the lines. Kay’s reviews of our manuscripts, theses, proposals, and presentations were masterclasses in effectively communicating science. The power of her red pen taught us scientific writing and the importance of constructive criticism. Her extensive edits, suggestions, and red-inked drafts inevitably made the next iteration better and conveyed to us lessons in scientific communication that many of us only appreciated later in our careers.

Kay has always been extremely generous with her time, knowledge, and reagents. One former post-doc remembers that Kay spent entire car trips between Bethesda and the USAMRIID facility, ensuring that he was grounded in virology so that he could make the transition from a cell biologist to a virologist. In many cases, these efforts instilled generosity and curiosity in her students. During the initial SARS-CoV pandemic, her entire group raced to understand this new virus but first worked collaboratively and collectively to generate and share reagents, cell lines, ideas, and data with other researchers with the same goal. Kay’s trainees took concepts, projects, and reagents to their new positions but also brought skills needed for effective collaborations.

Perhaps one measure of Kay’s impact is the diversity of contributions we, her mentees, have made. We are studying infectious diseases, the immune system, and the microbiome. We have written articles, essays, and books. We are researchers and professors involved in policy-making and teaching undergraduates. Many of us are working to understand SARS-CoV-2 as well as develop therapeutics and vaccines to combat this latest coronavirus pandemic. Despite the varied paths we have taken and the different roles we now fill, we all share Kay’s passion for science and her joy in sharing that passion with others.

## Figures and Tables

**Figure 1 viruses-14-01573-f001:**
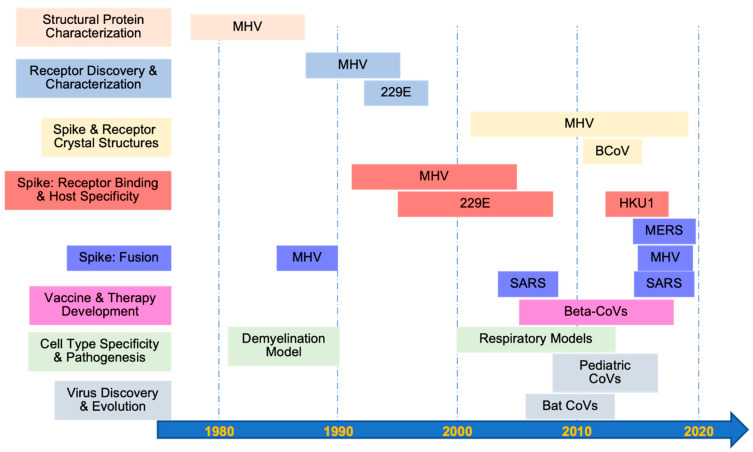
**Contributions to the coronavirus field from Dr. Kay Holmes’ laboratory group.** Areas of study are color coded by topics listed on the left for the specified coronaviruses shown in reference to the timeline across the bottom. MHV: mouse hepatitis virus; SARS: severe acute respiratory syndrome coronavirus; Beta-CoVs: betacoronaviruses; 229E: human coronavirus 229E; BCoV: bovine coronavirus; HKU1: human coronavirus HKU1; MERS: Middle East respiratory syndrome coronavirus; CoVs: coronaviruses.

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
