# Peer review of "Kathryn V. Holmes: A Career of Contributions to the Coronavirus Field"

_viruses, 2022, doi:10.3390/v14071573_

Round 1

Reviewer 1 Report

The extensive and fruitful research career of Kathryn Holmes is truly inspiring. Kay and her coauthors made discoveries that shaped many of our fundamental knowledge of coronaviral virology. The concepts and theories on receptor usage and specificity, which her team later proved to be accurate, formulated during her research are the foundation of our current understanding. Her work on the determination of receptors of MHV, 229E, and SARS-CoV are important contributions. The discoveries on spike-receptor interactions and cell-type specificity also provided novel insights. It is also a pleasure to read that Kay’s life work is not only on science but on nurturing the people doing it.

Here are some minor suggestions for edit:

1. Line 260 include citations for the discoveries for NL63 and HKU1 (e.g., 10.1038/nm1024, 10.1073/pnas.0400762101, 10.1128/JVI.79.2.884-895.2005)

2. Line 267 include citations for the view that Asian bats are important reservoir hosts for emerging viruses (e.g., 10.1073/pnas.0506735102, 10.1016/j.virol.2006.02.041, 10.1128/JVI.06540-11, 10.1128/CMR.00017-06)

Author Response

Thank you for your review. We have made the following changes in response to the reviewer's suggestions.

  1. Line 260 include citations for the discoveries for NL63 and HKU1 (e.g., 10.1038/nm1024, 10.1073/pnas.0400762101, 10.1128/JVI.79.2.884-895.2005)

These have been added, now line 354 in the marked manuscript.

  1. Line 267 include citations for the view that Asian bats are important reservoir hosts for emerging viruses (e.g., 10.1073/pnas.0506735102, 10.1016/j.virol.2006.02.041, 10.1128/JVI.06540-11, 10.1128/CMR.00017-06)

These have been added, now line 374 in the marked manuscript.

Reviewer 2 Report

In this manuscript, Bonavia et al describe Dr. Kathryn Holmes contributions to science and to mentoring. Dr. Holmes is an important figure in coronavirus research and is a well known supporter of young faculty. The manuscript is well written and describes her many contributions in sufficient detail. The authors do an excellent job describing Dr. Holmes’ mentoring skills.

One suggestion for improvement is that the authors spend more time discussing research contributions in which her laboratory was the clear pioneer (e.g., discovery of the murine coronavirus and HCoV-229E host cell receptors) from others in which she played a role, but her laboratory’s role was part of a larger group (e.g., studies of polarized cells). This would put emphasis on the major and unique contributions that she made to the field. A similar comment applies to Figure 1, in which major and less major contributions are not distinguished.

A minor suggestion is that the manuscript be carefully reviewed for errors in tense usage. The present and past tenses are used inconsistently throughout the manuscript.

Author Response

One suggestion for improvement is that the authors spend more time discussing research contributions in which her laboratory was the clear pioneer (e.g., discovery of the murine coronavirus and HCoV-229E host cell receptors) from others in which she played a role, but her laboratory’s role was part of a larger group (e.g., studies of polarized cells). This would put emphasis on the major and unique contributions that she made to the field. A similar comment applies to Figure 1, in which major and less major contributions are not distinguished.

Thank you for the suggestion. While we wanted to include the breadth of coronavirus research that Kay contributed to through-out her career, we put the majority of the focus on the discoveries of the MHV and 229E receptors and studies of spike/receptor interactions that were the largest focus Kay’s career. We also wanted to recognize the productive collaborations that Kay contributed to within the coronavirus field. Some information describing the finding of CD209L as a SARS-CoV receptor has been scaled back and additional clarification of collaborative work has been included. The new abstract places focus her major contributions. Figure 1 has been updated to remove minor contributions and the legend has been updated to reflect the changes and remove the suggestion that they were all key findings.

A minor suggestion is that the manuscript be carefully reviewed for errors in tense usage. The present and past tenses are used inconsistently throughout the manuscript.

This has been addressed throughout the manuscript.

Reviewer 3 Report

This is a wonderful summary of Kay Holmes contributions to coronavirology.  It was a pleasure to read.  This is history worth remembering. 

The authors, Kay's former students and postdocs, should be commended for crafting such a clear, thorough, honest and inspiring summary.  

Author Response

Thank you for your support, we enjoyed writing this article.